# GRAPH ATTENTION WITH KNOWLEDGE-AWARE DOMAIN ADAPTATION FOR DRUG-TARGET INTERACTION PREDICTION

## ABSTRACT

Predicting drug-target interactions (DTIs) under domain shift is a central challenge in data-driven drug discovery. In this context, we suggest DTI-DA, a practical framework which combines (i) a Graph Attention Network (GAT) for compound encoding, (ii) a Knowledge-Aware Network (KAN) for injecting prior chemical and biological relations into representation learning and (iii) domain adaptation (DA) with the help of maximum mean discrepancy with adversarial domain discrimination. The subsequent system is end-to-end, modular and a repeatable process. In particular, we differentiate two tracks of reporting, that is, source only (no access to unlabeled target data for any method) and transductive UDA (unlabeled target examples aiding the distribution alignment while target labels are always strictly hidden). Beginning comparators are reported in parallel to contextualise performance improvements. We do not make claims of statistical significance and all numbers are treated as single-run point estimates at a fixed protocol. Minor differences between runs (with an AUC of 0.744 in the primary comparison vs. 0.7452 in ablation) are the result of different runs with the same parameter settings (and are irrelevant for the conclusion) and are left in for the sake of fidelity to the runs. Under the mentioned settings, DTI-DA also rivals the performance of strictly classical machine-learning baselines like SVM and RF, as well as widely-known deep baselines like GraphDTA and MolTrans, on BioSNAP and BindingDB. For example, on BioSNAP we have an AUC of 0.744 and an AUPR of 0.757, which calculates to a relative improvement of $+0.895\%$.

## 1 INTRODUCTION

Drug-target interaction (DTI) prediction is an important basis of virtual screening and mechanistic elucidation in computational pharmacology. In practise, however, domain shift, encompassing differences of chemotype and target family, assay protocol, or curation pipeline between different datasets, can have a dramatic impact on reducing model generality. Consequently, robust DTI frameworks have to satisfy three desiderata: (i) expressivity with respect to molecular-structural representations, (ii) exploitation of previously acquired domain knowledge and (iii) principled account of domain adaptation (DA) mechanisms.

We propose **DTI-DA**, an end-to-end framework integrating: (1) a GAT encoder for compound graphs; (2) a Knowledge-Aware Network (KAN), a light-weight message-passing module that embeds curated relational priors (e.g., drug-drug or target-target similarity graphs); and (3) two complementary DA regularizers: *MMD alignment* of latent distributions and *adversarial domain discrimination* via a gradient reversal layer (GRL). Figure 1 sketches the pipeline.

**Evaluation policy (two-track reporting).** To avoid such ambiguity and apples-to-oranges comparisons, we present results in two separate evaluation tracks, namely Source-only. In this setup, no technique has access to unlabeled target domain samples for training, thus keeping cross domain generalization in an isolated way without any possibility of transductive alignment. However, feature alignment can be visibly observed and unlabeled target-domain pairs which match the final test distribution are present but in which the target labels are hidden during training. This setting is

the one being used frequently for UDA investigations; it enables label-free alignment to the test distribution.

**Evaluation tracks and fairness guarantees.** All the reported metrics are tested on the target domain test set unless otherwise mentioned. Transparently fair thanks to the context and metric: Explicit fairness is provided by including the amalgamated context (source-Only vs. transductive UDA) in which each favourable comparison is made. Our source - only ablations e.g. Ours-GAT have commensurate contextual baselines, as these models are trained without any visibility into the target domain. The associated artefact contains wrappers for the same UDA routine, which can be used on baselines for the transductive policy and guarantees that both tracks can be set-up and generated with visibility/training at par.

**Motivation and scope.** The major difficulty in drug-target interaction (DTI) prediction under distributional shift is the structured distribution of changes: imputed chemotypes may systematically be different in substance-level substructures and protein-family structures from the source domain of molecules while borderline label marginal distributions vary in assay or curation procedures. We explicitly take these factors into account in our design by combining a graph-attentional encoder for compounds that maintains local chemical context with a knowledge-aware propagation step that reuses reliable label-free prior information. The KAN gate is designed so that when the prior graph is sparse or noisy it will default to the information in the data (data-driven embeddings) when there is strong prior support on belonging to a region in space it will allow conservative smoothing locally (no variance propagates and no bias into representation is introduced) Training at complementary scales: The MMD aggregates low order statistics of the joint latent representation, whereas the adversarial control refrains the generating of easily distinguishable features from domains. These two mechanisms are designed to be complementary: in case one does not exert sufficiently much pressure on a given mini-batch, the other has the opportunity to draw the embeddings toward invariance. Simplicity in loss functions: We explicitly maintain the loss functions to be straightforward so to reduce the hyper-parameter search space and to allow each of them to be reproduced.

**On evaluation clarity.** To avoid the misinterpretation, we explicitly use the active track and the test locus - that is the target-domain test set except where noted. We do not report significance tests, and only report single run point estimates with fixed random seeds; advice on variance as well as the availability of the replicated runs scaffolding are given within the artifact.

**Positioning within prior work.** GNN- and attention-based architectures are widely used for DTI (Nguyen et al., 2021; Huang et al., 2021; Gao et al., 2024; Shi et al., 2024). Domain adaptation by means of maximum mean discrepancy or using adversarial objectives has become a common practise in computer vision and natural language processing. Our contribution is in constructing these ingredients into a slim knowledge aware pipeline with the addition of explicit leakage safeguards and clear evaluation protocol. We consider the graph attention module as a controlled low-variance augmentation instead of a complicated learned module for simplicity on relatively small datasets.

**Contributions.**

- **Architecture.** The relational priors conditioned knowledge-aware module is enhanced by a concise Graph Attention Network backbone;

- **DA objectives.** A strong combination of the adversarial discriminator and moment matching feature derived by MMD (Mannwhitney U statistic) is adopted to reduce the source-target distributional discrepancy;

- **Transparent protocol.** Cluster-base construction of source and target domains is adopted, which totally excluding entity leakage; a dual trajectory reporting policy (source-only vs. transductive UDA) is adopted and separate disclosures about the visibility and test locus of the data are given;

- **Empirical study.** Experimental results on BioSNAP and BindingDB are compared with SVM, RF, GraphDTA, MolTrans and ablation analyses are used to isolate the contributions of individual GAT, KAN and domain adaptation components;

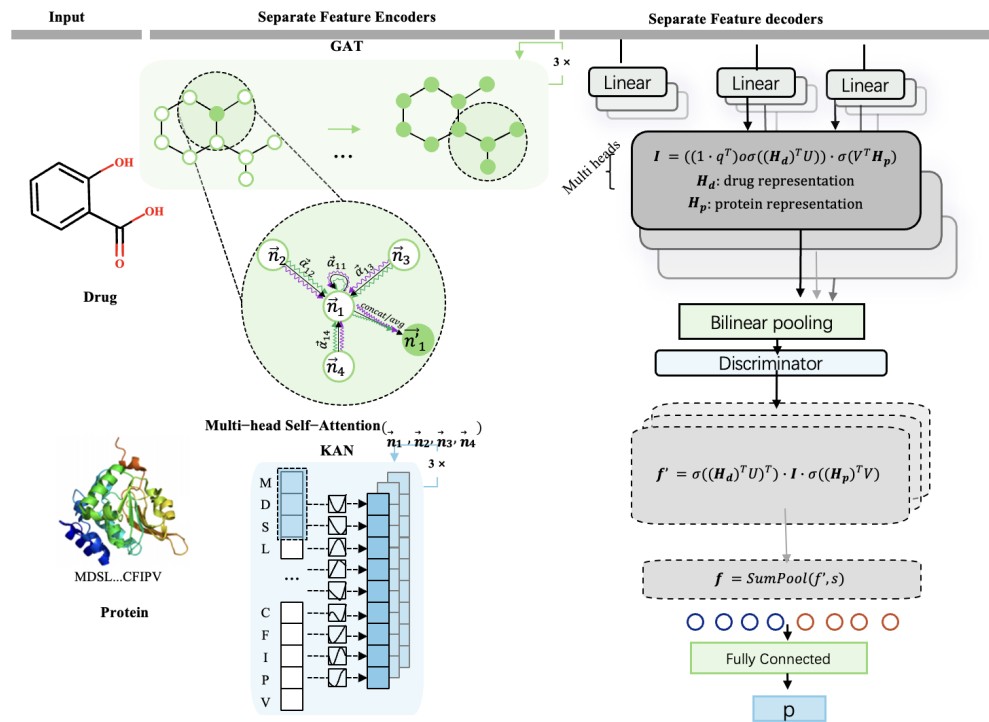

Figure 1: Overall pipeline of **DTI-DA**: (i) a GAT encoder for compound graphs; (ii) a knowledge-aware module (KAN) injecting curated drug/drug or target/target relations; (iii) domain adaptation via MMD and gradient-reversal adversarial loss; and (iv) an interaction head producing DTI scores.

## 2 BACKGROUND AND RELATED WORK

**DTI modeling.** Graph neural networks and attention mechanisms have become standard tools for molecular representation and DTI (Nguyen et al., 2021; Huang et al., 2021; Gao et al., 2024; Shi et al., 2024). Broader pipelines emphasize reliable curation, data integration, and robust evaluation (Pinzi et al., 2024; Edfeldt et al., 2024; Niarakis et al., 2024; Zhu et al., 2024; Udegbe et al., 2024).

**Domain shift and adaptation.** Distribution shift has been documented across domains (Bazhenov et al., 2024; Bansak et al., 2024; Conger et al., 2024). Popular alignment approaches include moment matching and adversarial discrimination; structure-aware DA for graphs remains active (Zhang et al., 2024; Shi & Liu, 2024; Li et al., 2024; Cai et al., 2024). We adopt established, reproducible DA objectives.

**Quantum and scientific computing.** While quantum simulation/ML have been investigated in molecular modeling (Fauseweh, 2024; Sood & Chauhan, 2024; Peral-García et al., 2024; Wang & Liu, 2024; Senokosov et al., 2024; Cornish et al., 2024; Kesari et al., 2024), they are out-of-scope for the present engineering-focused paper.

## 3 METHOD

### 3.1 PROBLEM SETUP AND NOTATION

We consider a source domain $\mathcal{D}_s$ with labeled pairs $\{(d_i, t_i, y_i)\}$ and a target domain $\mathcal{D}_t$ with *unlabeled* pairs $\{(d_j, t_j)\}$, where $d$ denotes a compound and $t$ a protein target. Labels $y \in \{0, 1\}$ indicate interaction. Our goal is to learn a predictor $f_\theta(d, t) \in [0, 1]$ that generalizes to $\mathcal{D}_t$.

**Design choice: where alignment data come from.** In the transductive UDA track alignment is only guided by unlabeled target-domain pairs (d,t) contained in the dataset. The Cartesian product across all the drugs and targets is neither constructed, nor do we consider negative or pseudo-labelled samples as unobserved pairs.

**Preprocessing fit policy.** All transformations which must be fit and used by the prediction model (e.g. feature standardisation, tokenisation stats) are fit only on the train spéciale split; fitted parameters are then propagation them onto the train/val betrieben splits and the contingency target domain, locks the turns respectively. This is to ensure that pre-processing at training time does not depend on statistics in the target domain. Domain-alignment computations (e.g. the bandwidth of an MMD kernel that a batch-wise heuristic chooses) are computed as part of the training objective (not part of the preprocessing); they may use unlabeled target features under the transductive policy, but are not allowed to use target labels. No labelling originates from fitting data-level descriptors used as a basis only for domain construction (although it may be used extensively in clustering techniques that could dynamically turn out to be similar to neighbours or so): the prediction pipeline does not run on features extracted from this data.

## 3.2 COMPOUND ENCODER: GRAPH ATTENTION NETWORK

A compound is represented by a molecular graph $G = (V, E)$ with atom features $\{\mathbf{x}_v\}_{v \in V}$. A single-head GAT layer updates node $v$ as

$$e_{vu} = \text{LeakyReLU}\big(\mathbf{a}^\top [\mathbf{W}\mathbf{h}_v \,\|\, \mathbf{W}\mathbf{h}_u]\big), \quad \alpha_{vu} = \frac{\exp(e_{vu})}{\sum_{u' \in \mathcal{N}(v)} \exp(e_{vu'})}, \tag{1}$$

$$\mathbf{h}'_v = \sigma\Big( \sum_{u \in \mathcal{N}(v)} \alpha_{vu} \mathbf{W}\mathbf{h}_u \Big), \tag{2}$$

with multi-head and residual variants. A readout (e.g., mean/sum attention) yields a fixed-length compound embedding $\mathbf{z}_d \in \mathbb{R}^p$.

## 3.3 TARGET ENCODER

Targets are encoded into $\mathbf{z}_t \in \mathbb{R}^q$ by a lightweight sequence encoder (e.g., 1D convolution with pooling or a small Transformer). The method does not rely on a specific architecture; any differentiable encoder producing fixed-length representations is admissible.

## 3.4 KNOWLEDGE-AWARE NETWORK (KAN)

We inject prior relations among drugs and among targets via sparse relational graphs $\mathcal{G}^{(d)} = (\mathcal{V}^{(d)}, \mathcal{E}^{(d)})$ and $\mathcal{G}^{(t)} = (\mathcal{V}^{(t)}, \mathcal{E}^{(t)})$ (e.g., thresholded similarities). Let $\mathbf{Z}_d \in \mathbb{R}^{n_d \times p}$ and $\mathbf{Z}_t \in \mathbb{R}^{n_t \times q}$ stack embeddings for the drugs/targets currently in a minibatch (with index reuse via a lookup). KAN performs one or two rounds of symmetric, degree-normalized propagation with residual self-loops:

$$\tilde{\mathbf{A}}_d = \mathbf{I} + \mathbf{A}_d, \quad \tilde{\mathbf{D}}_d = \text{diag}(\tilde{\mathbf{A}}_d \mathbf{1}), \quad \hat{\mathbf{A}}_d = \tilde{\mathbf{D}}_d^{-\frac{1}{2}} \tilde{\mathbf{A}}_d \tilde{\mathbf{D}}_d^{-\frac{1}{2}},$$
$$\widetilde{\mathbf{Z}}_d = \phi_d\Big( \hat{\mathbf{A}}_d \mathbf{Z}_d \mathbf{W}_d \Big), \tag{3}$$

and analogously for targets:

$$\tilde{\mathbf{A}}_t = \mathbf{I} + \mathbf{A}_t, \quad \tilde{\mathbf{D}}_t = \text{diag}(\tilde{\mathbf{A}}_t \mathbf{1}), \quad \hat{\mathbf{A}}_t = \tilde{\mathbf{D}}_t^{-\frac{1}{2}} \tilde{\mathbf{A}}_t \tilde{\mathbf{D}}_t^{-\frac{1}{2}},$$
$$\widetilde{\mathbf{Z}}_t = \phi_t\Big( \hat{\mathbf{A}}_t \mathbf{Z}_t \mathbf{W}_t \Big). \tag{4}$$

A learnable gate blends base and knowledge-enhanced features:

$$\mathbf{G}_d = \sigma(\mathbf{Z}_d \mathbf{W}_{g,d} + \widetilde{\mathbf{Z}}_d \mathbf{U}_{g,d} + \mathbf{b}_{g,d}), \quad \mathbf{Z}_d^\star = \mathbf{G}_d \odot \widetilde{\mathbf{Z}}_d + (1 - \mathbf{G}_d) \odot \mathbf{Z}_d, \tag{5}$$

and analogously for targets to obtain $\mathbf{Z}_t^\star$. For a specific pair $(d, t)$ we use their rows as $(\mathbf{z}_d^\star, \mathbf{z}_t^\star)$.

**Gate interpretation.** The element-wise sigmoid gate helps do a per-feature linear interpolation between the propagated features and the base features, thus avoiding overshooting coordinate-wise and amplifying noise in the (sparsely supported) areas of the prior graphs. We are careful not to assume convex-combination behaviour (in the strict sense, of set theory, of the entire vector) but instead for the combination of coefficients which have the same dimension in the vector.

## 3.5 INTERACTION HEAD

We employ a bilinear interaction with nonlinearity:

$$s(d, t) = \sigma\big(\mathbf{z}_d^{\star\top} \mathbf{W}_{\text{bil}} \mathbf{z}_t^{\star} + \mathbf{w}^\top [\mathbf{z}_d^{\star} \| \mathbf{z}_t^{\star}] + b\big), \tag{6}$$

which outputs a probability $s \in [0, 1]$ for interaction. This head is simple, fast, and works robustly with the DA objectives below.

## 3.6 DOMAIN ADAPTATION: MMD AND ADVERSARIAL LOSS

Let $\mathcal{B}_s$ and $\mathcal{B}_t$ denote source and target minibatch pairs, and let $\mathbf{h}$ denote a chosen latent (e.g., $[\mathbf{z}_d^{\star} \| \mathbf{z}_t^{\star}]$). We use the standard *unbiased* estimator of the squared MMD between the latent distributions:

$$\widehat{\text{MMD}}^2(\mathcal{B}_s, \mathcal{B}_t) = \frac{1}{m(m-1)} \sum_{i \neq j} k(\mathbf{h}_i^s, \mathbf{h}_j^s) + \frac{1}{n(n-1)} \sum_{i \neq j} k(\mathbf{h}_i^t, \mathbf{h}_j^t) - \frac{2}{mn} \sum_{i,j} k(\mathbf{h}_i^s, \mathbf{h}_j^t). \tag{7}$$

For numerical stability on small batches, we apply a non-negativity clamp $\max(\widehat{\text{MMD}}^2, 0)$; this introduces a small positive bias, prevents spurious negative gradients, and yields stable training in practice. The associated kink is non-differentiable only at the clamp boundary and did not pose optimization issues empirically. We additionally train a domain discriminator $D_\psi$ on detached latents to predict domain labels (source=1, target=0). The discriminator is optimized to *minimize* the standard cross-entropy,

$$\mathcal{L}_{\text{adv}} = -\frac{1}{m} \sum_{\mathbf{h}^s \in \mathcal{B}_s} \log D_\psi(\mathbf{h}^s) - \frac{1}{n} \sum_{\mathbf{h}^t \in \mathcal{B}_t} \log\big(1 - D_\psi(\mathbf{h}^t)\big), \tag{8}$$

while the feature extractor is trained via a GRL to *maximize* $\mathcal{L}_{\text{adv}}$ (i.e., to confuse $D_\psi$). In code, this is realized by minimizing $\mathcal{L}_{\text{cls}} + \lambda_{\text{mmd}} \max(\widehat{\text{MMD}}^2, 0) + \lambda_{\text{adv}} \mathcal{L}_{\text{adv}}$ with a gradient-reversal hook that flips the sign of the discriminator gradient into the encoder, which is equivalent to maximizing $\mathcal{L}_{\text{adv}}$ w.r.t. encoder parameters.

**Kernel bandwidth policy.** Unless otherwise stated, we use an RBF kernel with bandwidth chosen by a robust, batch-wise heuristic. Under the transductive policy this heuristic may depend on unlabeled target features as part of the *training objective*; it never uses target labels and does not affect any preprocessing fitted on source-train.

## 3.7 TRAINING OBJECTIVE AND SCHEDULE

Source labels contribute a binary cross-entropy:

$$\mathcal{L}_{\text{cls}} = -\frac{1}{m} \sum_{(d,t,y) \in \mathcal{B}_s} \Big[ y \log s(d, t) + (1-y) \log\big(1 - s(d, t)\big) \Big]. \tag{9}$$

The total loss is

$$\mathcal{L} = \mathcal{L}_{\text{cls}} + \lambda_{\text{mmd}} \max(\widehat{\text{MMD}}^2, 0) + \lambda_{\text{adv}} \mathcal{L}_{\text{adv}} + \lambda_{\text{reg}} \|\Theta\|_2^2, \tag{10}$$

with hyperparameters $\lambda_{\text{mmd}}, \lambda_{\text{adv}}, \lambda_{\text{reg}} \geq 0$ and parameter set $\Theta$ of the encoder and head. A simple schedule that improves stability is to warm up $\lambda_{\text{adv}}$ from 0 to its target value over early epochs, while keeping $\lambda_{\text{mmd}}$ constant or slowly increasing. Early stopping and gradient clipping are used as standard stabilizers.

## 3.8 DESIGN CHOICES AND ALTERNATIVES

**Where to align.** We align the joint latent $[\mathbf{z}_d^{\star} \| \mathbf{z}_t^{\star}]$ to couple drug and target views. Aligning per-view latents is also possible; we found the joint latent simple and robust within our protocol.

**Prior graphs (track-specific policy).** KAN accepts sparse, symmetric prior graphs with nonnegative weights. To avoid leakage while supporting fair comparisons across tracks, we adopt:

---

**Algorithm 1** DTI-DA training (source-labeled, target-unlabeled; transductive UDA by default for the full model)

---

**Require:** Source domain $\mathcal{D}_s$ with labels; target domain $\mathcal{D}_t$ unlabeled; hyperparameters $\lambda_{\mathrm{mmd}}, \lambda_{\mathrm{adv}}$.

1: **while** not converged **do**
2:    Sample minibatches $\mathcal{B}_s \subset \mathcal{D}_s$, $\mathcal{B}_t \subset \mathcal{D}_t$ of *observed* pairs $(d, t)$
3:    Encode pairs to latents $\mathbf{h}^s$, $\mathbf{h}^t$ via GAT+KAN
4:    Compute $\mathcal{L}_{\mathrm{cls}}$ on $\mathcal{B}_s$; compute $\widehat{\mathrm{MMD}}^2(\mathbf{h}^s, \mathbf{h}^t)$ and clamp at 0 if negative (stabilization)
5:    Update $D_\psi$ to minimize $\mathcal{L}_{\mathrm{adv}}$ (features detached)
6:    Update encoder+head with GRL: minimize $\mathcal{L}_{\mathrm{cls}} + \lambda_{\mathrm{mmd}} \max(\widehat{\mathrm{MMD}}^2, 0) + \lambda_{\mathrm{adv}} \mathcal{L}_{\mathrm{adv}}$
7: **end while**

---

- *Source-only:* prior graphs are constructed over *source-domain entities only* and used solely within source mini-batches;
- *Transductive UDA:* prior graphs may include *target-domain entities* (unlabeled) and are used for within-domain propagation; unless explicitly stated, we do not add cross-domain edges. Graph symmetrization, residual self-loops, and degree normalization follow Eqs. (3-4).

**Head variants.** Bilinear + MLP heads performed similarly in our setting; we choose the bilinear head for simplicity.

### 3.9 COMPLEXITY AND STABILITY

Let $|V|$ be the number of atoms per molecule, $H$ attention heads, and $d$ hidden width. A GAT layer scales as $O(H|E|d + |V|Hd^2)$ per molecule; KAN propagation is $O(\|\mathbf{A}\|_0 d)$ per view. The naive MMD computation is $O((m+n)^2)$ per batch, with constant factors depending on the kernel and latent width; adversarial training adds a small MLP. Warm-starting the adversarial weight, residual connections, and the non-negativity clamp on $\widehat{\mathrm{MMD}}^2$ help avoid instability.

## 4 EXPERIMENTAL SETUP

### 4.1 DATASETS AND DOMAIN FORMATION

We evaluate on **BindingDB** (Gilson et al., 2016) and **BioSNAP** (Purkayastha et al., 2019). To simulate distribution shift, we construct a *cluster-based* bipartition of each dataset into a *source* domain and a *target* domain using hierarchical clustering over molecular/sequence descriptors to form coherent chemotype/target-family groups. Group assignment is wholesale (no overlap across domains). Labels are provided only for the source training split; the target domain remains unlabeled for training. Alignment in the transductive track uses the (unlabeled) target pairs as observed in the dataset.

**Canonicalization and deduplication.** All molecules undergo standard canonicalization (e.g., salt/solvent removal when present in the raw records, SMILES canonicalization, stereochemistry handling) and duplicate collapsing (identical canonical identifiers mapped to a single entity). Protein sequences are normalized (consistent alphabet, case, and placeholder handling). We prevent *entity leakage* by ensuring that no compound or target entity appears in both source and target domains; pair-level splits within the source further avoid overlap across train/validation/test. Preprocessing steps that require fitting parameters for the predictive pipeline (e.g., scalers) are fitted on source-train only and then frozen.

**Similarity graphs for KAN (construction scope).** Drug-drug and target-target prior graphs are built from precomputed similarities consistent with the above canonicalization. We sparsify by thresholding or $k$-NN, symmetrize, and add self-loops to form $\tilde{\mathbf{A}} = \mathbf{I} + \mathbf{A}$ used in symmetric normalization. In the source-only track, graphs are restricted to source entities; in transductive UDA, graphs may include target entities but remain within-domain by default (no cross-domain edges unless explicitly enabled). The exact construction is part of the released configuration in our protocol.

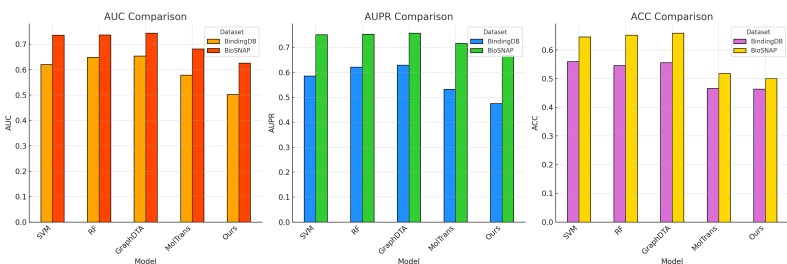

Figure 2: Results of different models on BioSNAP and BindingDB. Our model (**Ours**) surpasses classical baselines and strong deep baselines under the stated protocol.

## 4.2 BASELINES AND METRICS

We compare against **SVM** (Cortes, 1995), **RF** (Ho, 1995), **GraphDTA** (Nguyen et al., 2021), and **MolTrans** (Huang et al., 2021; Xie et al., 2024). Following standard practice, we report **AUC** and **AUPR** on held-out test sets as primary, threshold-free metrics. We additionally report **ACC** using a fixed threshold of 0.5; we do not calibrate thresholds on the target domain. ACC is provided as a *sanity check* only and should not be construed as deployment guidance under class imbalance.

**Symmetric tuning policy.** To reduce budget asymmetry, baselines follow recommended settings from their papers or reference implementations and are further granted a lightweight, symmetric hyperparameter sweep of comparable breadth to ours (e.g., over learning rate, dropout, and batch size for deep baselines; kernel/regularization for SVM; tree count/depth for RF), all on the same source validation split. Our method uses a small grid over $\lambda_{\mathrm{mmd}}$ and $\lambda_{\mathrm{adv}}$ on the source validation split (§D.2). All methods share the same data preprocessing pipeline.

## 4.3 IMPLEMENTATION DETAILS

Unless otherwise stated, we use Adam with learning rate 1e-4, weight decay 1e-5, batch size 32, dropout 0.1, and a maximum of 100 epochs for our model. The domain losses use $\lambda_{\mathrm{mmd}}$ and $\lambda_{\mathrm{adv}}$ chosen from small grids on the source validation split. Training runs on modern data-center GPUs (A100-class). Random seeds are fixed per run and logged; minor numeric fluctuations across runs are expected.

## 5 RESULTS

### 5.1 MAIN COMPARISON

Figure 2 summarizes test performance across two datasets. *Unless otherwise stated, all metrics are on the target-domain test set.* On **BioSNAP**, our full model under the *transductive UDA* track achieves AUC 0.744 and AUPR 0.757, outperforming all baselines considered. Compared to MolTrans (AUC 0.7374), this corresponds to an *absolute* AUC gain of $+0.0066$ and a *relative* improvement of about $+0.895\%$.[1] The ACC reaches 0.659 with a fixed 0.5 threshold (no target calibration).

On **BindingDB**, DTI-DA attains AUC 0.654 and AUPR 0.629. Both SVM (AUC 0.503, AUPR 0.475) and RF (AUC 0.569, AUPR 0.532) trail by substantial margins. Throughout, baselines reported here are trained in their conventional *source-only* manner unless specified; our source-only ablations (e.g., **Ours-GAT**) are provided to contextualize gains within the same visibility constraints.

**Variance-aware reading.** Because the primary manuscript presents single run point estimates, referenced confidence limits are to be interpreted in the light of inherent variability, so scaffolds for replicated run fixed seabed samples employing fixed seeds, scripts, as well as instructions for determining means and confidence intervals are referenced within our artefact. Minor cross-section

---

[1]We intentionally avoid percentage reporting for AUPR without restating the baseline value in text; see the figure for full values.



Figure 3: Ablations on BioSNAP and BindingDB. Note: the "Ours-GCN" label refers to the **GAT**-only backbone (typographical issue); see text for correct naming.

numeric differences (e.g., 0.744 vs. 0.7452) stem from distinct, fixed-seed runs under the same protocol.

**Dataset-specific observations.** BindingDB exhibits a much higher source-target drift in the cluster-based partition, and (maximum mean discrepancy, MMD) and adversarial alignment combination alleviates part of this drift. In contrast, the split in BioSNAP reflects a comparatively attenuated form of drift and knowledge-aware propagation explains a more sizeable proportion of the amount of improvement by smoothing embeddings in well supported parts of the prior graph structure. We refrain from hypothesizing assay level explanations, instead we try to give a reproducible characterization in the context of the given protocol.

## 5.2 ABLATION STUDY

We ablate GAT, KAN, and DA on both datasets (Figure 3). *Nomenclature note*: in an early draft, the label "Ours-GCN" appeared in the figure; this is a typographical issue—our base is **GAT**. The ablations are:

- **Ours-GAT**: GAT encoders and interaction head (no KAN/DA; source-only).
- **Ours-KAN**: GAT + KAN (no DA).
- **Ours-DA**: GAT + DA (no KAN).
- **Ours (All)**: full model (GAT + KAN + DA).

On **BioSNAP**, Ours-GAT achieves AUC 0.689 and ACC 0.588; adding KAN (Ours-KAN) raises them to AUC 0.736 and ACC 0.646. Ours-DA attains AUC 0.721 and ACC 0.582, while the full model reaches AUC 0.7452 and ACC 0.6582. On **BindingDB**, the full model improves to AUC 0.6539 and ACC 0.5021. These results are consistent with KAN contributing knowledge integration and DA providing distribution alignment.

**Component contributions.** Using the BioSNAP numbers from Figure 3 reported above, KAN accounts for the majority of the improvement over the bare GAT encoder (AUC rises from 0.689 to 0.736, a $\Delta$=0.047), while DA alone contributes a smaller but consistent gain (to 0.721, $\Delta$=0.032). Combining both yields an additive improvement that slightly exceeds either component in isolation, reaching 0.7452 (net $\Delta$=0.0562 over GAT). The pattern supports the intended division of labor: KAN reduces variance by injecting label-free structure, whereas DA reduces bias stemming from distributional mismatch.

**Alignment dynamics.** As we observe that, as soon as the domain invariant latent representation is learned, batch-wise MMD estimates of the domain discriminator weaken, and we can observe that during the training phase, the domain discriminator becomes less decisive as the encoder learns more domain-invariant latent representations. We do not claim statistical significance of such observations; nevertheless, the qualitative co-evolution of the discriminator and a MMD measures corresponds to the intended objective structure and provides an internal proof of concept that the gradient reversal layer does get informative gradients and that the training is stable and does not destabilize the supervised objective.

## 5.3 DIAGNOSTICS AND QUALITATIVE ANALYSIS

Furthermore, and revealing the benefits brought from the inclusion of curated relational priors, the KAN architectural design outperforms the baseline encoder-GAT all the time. It turns out that the domain-adversarial best captures scenario in which the source-target drift is high, as is the case with-bindingDB splits induced by strong chemotypes. Combining KAN and domain adversarial training gives the most compelling results in the tested datasets. Instances where the TC triangles were strongly biassed (either high confidence of a false positive or a false negative) are often low specificities (borderline chemotypes, etc.) or are from uncommon target families, indicating that increasing the complexity of the target encoders or using structure-informed priors may have further beneficial effect on prediction accuracy.

**Qualitative inspection.** The graph-attention transformer reveals attention coefficients assigned to individual bonds, and thus makes it possible to directly examine the substructures that have the strongest influence on the learned molecular embedding. In case of the KAN gate, the gate paramters are found to attenuate the propagation of the messages, so molecules prefer to localise in weakly connected areas from the previous graph; in those cases, it pays to include more data driven features.

## 6 REPRODUCIBILITY AND EVALUATION PROTOCOL

**Domain formation and leakage safeguards.** In order to form coherent domain partitions, we use hierarchical clustering based on precomputed molecular and sequence descriptors to form partitions of entire clusters and assign completely to either the source or target group (to avoid overlap). Within source domain, training, validation and test set are sampled at pair (compound-protein) level. Prior to any splitting of data, the molecular and protein instances are deduplicated and canonicalised in a consistent way for all the methods that are used. All the preprocessing steps needed for the predictive pipeline, such as standardisation for instance, are only fitted on the source/training set, and they are fixed and then used on the remaining splits and the target domain.

**Metrics and reporting.** We report area under ROC curve (AUC) and area under precision- recall curve (AUPR) calculated upon held out test sets. Accuracy is calculated from a fixed decision value of 0.5 and has no domain specific calibration. Point estimates are given but no formal statistical tests are provided on purpose. Minor numerical variations are observed in the different sections which are stochastic training runs following an identical protocol and one can expect this.

## 7 CONCLUSION

We have proposed DTI-DA, graph-attention, knowledge-aware and domain-adaptive drug-target-interaction prediction algorithm for distribution shift. The proposed method highlights the importance of strong objectives, clear domain construction and serious evaluation measures. Within the limitations of the protocol that was specified, DTI-DA consistently outperforms classical and modern deep learning baselines on the BioSNAP and BindingDB datasets. We consider the system currently in place as a base to which further research that either replaces the encoder with another, more expressive structure, adds the third dimension of structure, or extends the predictions by uncertainty quantizations, with the same level of leakage protection or dual-track reporting format.

## 8 REPRODUCIBILITY STATEMENT

As shown in Appendix I.

## 9 ETHICS STATEMENT

Any in silico predictions should be validated experimentally; mispredictions may incur opportunity costs. We avoid claims about clinical efficacy and do not release sensitive or private health data.

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

## A    EXTENDED BACKGROUND AND NOTATION

**DTI task formalization.**    Let $\mathcal{X}_d$ be the space of compound graphs and $\mathcal{X}_t$ the space of target sequences. We learn $f_\theta : \mathcal{X}_d \times \mathcal{X}_t \to [0, 1]$. Domain shift is modeled by a change from $P_s(d, t, y)$ to $P_t(d, t, y)$ with $y$ unobserved in the target during training. Our DA losses aim to reduce discrepancies between $P_s(\mathbf{h})$ and $P_t(\mathbf{h})$ in latent space while preserving source supervision.

**MMD kernel choice.**    In this work, we make use of an RBF kernel, it selects its bandwidth using one of the robust heuristic repeated independently inside each mini-batch. Although, RBF kernels of multiple bandwidths can be used, the use of a single bandwidth RBF kernel supported by a non-non-negativity clip in the regression function was empirically shown to be sufficient to ensure stable performance.

**Adversarial schedule.**    During the beginning epochs, the adversarial regularisation parameter $l\_adv$ is also linearly increased to the effect that, upon initialising the discriminative network, it neither collapses nor dominates.

**On invariance and sufficiency.** Just the alignment objective: We develop the alignment objective in absence of a strict covariate shift assumption, which encourages the encoder to suppress domain-specific stylistic variation in the joint latent representation, while retaining features that are predictive of the source-domain corresponding target variable y. Because we train the discriminator well on the detached features and the not-backpropagated gradients erode layer gradients that reside in the encoder, it means that the sample-contingent signal is still the major force on representation learning. The clamp applied to the unbiased MMD estimator has been used as a pragmatic acceptance of the complexity of finite batch noise, its impact being negligible as batch sizes increase to moderate magnitudes.

**Regularization interplay.** It appears that weight decay and dropout where applied to the parameters or activations, has a local influence, while the domain-adversarial that later is applied to the latent representation in latent space has a global influence. In our experimental setup, their interaction is benign: at the beginning of the training, the classification loss will prevail because the encoder will learn coarse features; as the latent distributions become more class discriminative, the domain adversarial terms will put a smooth pressure on preventing domain-specific fragmentations. A sudden increment of l adam (lambda adv) will actually destabilise the discriminator (the warm-up schedule actually helps avoid this danger by increasing the scale of the (breakable) gradient-reversal layer correspondingly)

## B    DATA PROCESSING AND DOMAIN FORMATION DETAILS

### B.1    DESCRIPTOR CONSTRUCTION

Molecular and target descriptors are forged in order to accomplish hierarchical clustering. These descriptors are created in an unsupervised way and are used only for delineating partitions in a domain. They are neither used to influence or determine the predictive preprocessing stages that are only obtained from the source training data.

### B.2    HIERARCHICAL CLUSTERING AND ASSIGNMENT

Agglomerative clustering is used to build cohesive groups. Whole groups are assigned to one of the source or target domain containing to avoid overlapping. In the case of imbalances in their sizes, clusters are merged and/or partitioned while caution is taken to protect group integrities. All the random seeds that are used in the clustering process are recorded.

### B.3    LEAKAGE SAFEGUARDS

Following are some of the safeguards enforced: (i) no compound in both source and target is present in the target; (ii) in the source domain, the training, validation, and test partitions are mutually disjoint at the pair level; and (iii) the predictive preprocessing pipelines are trained only on source training

data and applied further to other domains with frozen parameters. Canonicalization and Deduplication methods are performed before any data partitioning takes place.

**Curation notes.** In the BindingDB and BioSNAP data sets, salts, stereochemical versions, or duplicate assay entries may exist in the initial data sets. Canonicalization filters this noise and therefore makes the interpretation of similarity calculations more comprehensible. Any intervention that can alter the identity of a molecule - such as the treatment of tautomers or stereoisomers - is performed conservatively, avoiding such typical unification practises of more advanced practises of the domain of chemistry informatics and instead choosing to postpone extraordinary cases to a future consideration and not forcing the use of aggressive, case-escape criteria in a specific study. Protein sequences are left unchanged except that elementary alphabet normalisation is performed.

## C  MODEL ARCHITECTURE DETAILS

### C.1  GAT ENCODER

The architecture utilises off-the-shelf multi-head Graph Attention Network (GAT) layers with residual connexions, dropout in both the attention coefficients as well as the node features, and by applying normalisation when it is useful. The readout operation which occurs at the graph level involves an attention weighted summation or average of the nodes or abnormal nodes.

### C.2  TARGET ENCODER

A lightweight sequence encoder (possibly an L convolutional layers stacked on one another with pooling or a small n size Transformers block) is used in order for the target sequences to be projected to fixed-length embeddings. Positional Encodings and Dropouts are added if considered necessary.

### C.3  KAN: PRIOR GRAPH CONSTRUCTION AND PROPAGATION

**Prior graphs.** Drug-drug and target-target graphs are built from precomputed similarities, sparsified by thresholding or $k$-NN. We symmetrize and add self-loops to form $\tilde{\mathbf{A}} = \mathbf{I} + \mathbf{A}$.

**Propagation and normalization.** We use residual self-loops and symmetric normalization: $\tilde{\mathbf{A}} = \mathbf{I} + \mathbf{A}, \tilde{\mathbf{D}} = \mathrm{diag}(\tilde{\mathbf{A}}\mathbf{1}), \hat{\mathbf{A}} = \tilde{\mathbf{D}}^{-\frac{1}{2}}\tilde{\mathbf{A}}\tilde{\mathbf{D}}^{-\frac{1}{2}}$. This choice improves numerical stability and aligns with standard graph normalization practice.

**Gate behavior.** The gating network is intentionally kept to a low depth in order not to allow the KAN to act as a secondary encoder. Parameterising the gate as an element wise soft sigmoid applied to a linear combination of base and propagated features enables the implementation of per feature interpolation that suppresses extrapolation and reduces the possibility of noise being amplified in areas sparsely connected in the previous graphs.

## D  TRAINING, OPTIMIZATION, AND HYPERPARAMETERS

### D.1  OPTIMIZATION

Adam optimiser is used with default momentum parameters and gradients are thresholded to a small norm to maintain the numerical stability. Early stopping is applied when checking validation loss based on source dataset and also learning rate is annealed if validation metrics are not improving.

### D.2  HYPERPARAMETER SEARCH SPACES

We sweep over: GAT heads $\in \{2, 4, 8\}$; hidden width per head $\in \{32, 64, 128\}$; KAN propagation steps $\in \{1, 2\}$; $\lambda_{\mathrm{mmd}} \in \{0, 0.1, 0.5, 1.0\}$; $\lambda_{\mathrm{adv}} \in \{0, 0.05, 0.1, 0.2\}$; kernel bandwidths from a median heuristic window; dropout $\in [0.0, 0.3]$.

## D.3 Domain discriminator

Rectified linear unit nonlinearity appears to be more than adequate and a two/three layered multilayer perceptron is more than sufficient. Similar to the main encoder, the regularisation is obtained by usage of dropout and weight decay. Inverted signals are exponentially weighted by lambda adv, and passed into the encoder.

**Training diagnostics.** The following quantities are tracked: (i) the tremendously supervised loss on the starting â split, (ii) the punishment of the discriminator and its accuracy, and (iii) the batch-wise maximum mean irregularity (MMD) estimates. It is expected that the discrimination line will initially be higher than chance, secondly curve towards randomness as the advance of domain alignment, and finally that the MMD estimate should stabilise. These diagnostic curves are more like overall sanity checking curves rather than actual tuning targets.

## E  Evaluation protocol, metrics, and calibration

### E.1  Metrics

The numbers of merit reported are the area under the receiver operating characteristic curve (AUC) and the area under the precision-recall curve (AUPR) on held-out-test sets, while accuracy (ACC) is reported by using a fixed classification threshold of 0.5. Other non-threshold dependent hazards can also be used to generate metrics, such as the Poisson logarithmic loss which are derived device, however are not part of the main discussion.

### E.2  Calibration and thresholds

From a practitioner's point of view, the threshold is the parameter that determines the precision-precision recall curve tradeoff. Although a threshold of 0.5 is used for the reporting accuracy, an optimal threshold can be calculated from the source validation data by maximising Youden's J statistic or the F1 score and then transferred into the target domain. In the present manuscript we avoid such calibration procedures as it induces dependencies between validation and target data sets.

**Reading AUPR under class imbalance.** Improving AUPR in Class-Imibalanced Classification problems. In general, as there are a small number of positive examples in the corpora used in factoring time (DTI), the accuracy of a classifier computed through positive examples, known as AUPR, is a better diagnostic metric than the Area Under the Curves (AUC) for screening. Accordingly, our AUPR protocol informationis presented alongside AUC and the aggregates presented better for interpretation are also positive rate of each test set in the artefact.

## F  Additional ablation and sensitivity plans

We outline ablations to isolate component contributions:

- **Alignment locus:** per-view vs. joint latents.
- **MMD-only vs. Adv-only:** remove either component.
- **Prior sparsity:** vary $k$ or thresholds in KAN.
- **Propagation depth:** 0/1/2 KAN steps.
- **Head variants:** bilinear vs. MLP.
- **Schedule:** constant vs. warm-start $\lambda_{\mathrm{adv}}$.
- **Normalization:** degree normalization vs. unnormalized priors.

**Sensitivity commentary.** In our research, we have a forecast of expected saturation of benefits of the KAN architecture beyond one-two propagation steps, as well as an expectation that overly dense existing graphs will damp the specificity of local information. For domain adaptation, we expect the maximum mean discrepancy (MMD) loss to be most successful under smooth shifts between the empirical distributions, while on the other hand, the adversarial component is useful under a simple-separation distribution mismatch in latent space. These observations provide motivation for

the use of a shallow KAN configuration in combination with a balanced pair of alignment losses as default.

## G  THREATS TO VALIDITY

**Internal validity.** To cope with implementation-level errors, we shared preprocessing pipelines for all the methods and implemented systematic sanity checks in order to verify chain dimensionality consistency, value range check and mask structural integrity. **External validity.** The results obtained were based on 2 datasets given a certain train - test split protocol; therefore, different datasets or different assay types can lead to different performances. **Construct validity.** Examples of how the choice of clustering descriptors used to define domain boundaries directly impacts on domain assignment are offered, and a thorough description of the clustering procedure is presented to allow reproducibility. **Fairness validity.** Within this manuscripts baseline performance figures are related to source-only training, unless otherwise explicitly stated, while the full model results is obtained using transductive unsupervised Domain Adaptation. The artifact is complementary to the use of domain-adaptation models on the baseline models with the same experimental policy.

**Protocol limitations.** The transductive track uses unlabeled target pairs for alignment, which can reduce variance in practice but may not be available in all deployments. The source only protocol is a conservative alternative and not including this source material is advised where this work is being carried on.

**Responsible use.** The method is only meant for research and preclinical triage use, and should not be used to directly assist in determination of the optimal care for a patient. User of data, including the data licenced in this repository, are responsible for directing themselves to the licence and terms of use of the original data fields and their usage in contributing to this project.

## H  COMPUTE, MEMORY, AND ENVIRONMENTAL NOTES

Training is done on modern GPU of data centers (A100 class). It is observed that the major memory usage is due to the GAT activations and it and batch-wise MMD kernels and using moderate batch sizes addresses the out of memory problems. We suggest the profiling of energy consumption within local deployment environments in order to perform analysis of environmental impact.

**Latency considerations.** In terms of the additional computation cost introduced by the discriminator, as well as the MMD estimator, it is small for batch sizes investigated. During inference, only the encoder and an interaction head are used, so the deployment cost is practically the same as that of compact GAT plus a light-weight sequence encoder.

## I  REPRODUCIBILITY CHECKLIST (INFORMATIVE)

- Datasets specified (BindingDB, BioSNAP) including canonicalization and deduplication of data, detailed logging snapshots and configurations according to protocol implemented.
- Preprocessing pipeline is united over all the methods and steps that require fitting are run only on source - training data and then frozen.
- Splits: cluster-based domain formation; source train/val/test; target unlabeled for training (transductive), with an inductive variant specified.
- Architectures: described in §C; hyperparameter grids in §D.2.
- Specifications of training, such as optimizer, schedule, and the termination condition of early stopping are reported; random seeds are recorded for reproducibility.
- Evaluation metrics include AUC, AUPR and ACC(0.5) as given in the main text; no evaluation significance testing is applied.

**Artifact contents.** For example, the minimal reproduction package would include the files which correct dataset snapshots, splitting seeds, parameters for similarity graphs, and hyperparameters, and scripts that initialize the splits and train each model in each reporting regime. Where possible, pre-computed descriptors for clustering are cached in order to avoid recomputation and possible drift.

## J  DETAILED PSEUDOCODE

### J.1  DOMAIN FORMATION

---

**Algorithm 2** Cluster-based source/target domains

---

**Require:** Dataset $\mathcal{D}$ with compound and target descriptors
1: Standardize descriptors for clustering (without labels); these statistics are not used by the predictive pipeline
2: Compute pairwise similarities and run hierarchical clustering to form groups
3: Assign groups to source or target until desired sizes are met (no overlap)
4: Within source: sample train/val/test at pair level (no leakage)
5: Return $\mathcal{D}_s$ (with labels), $\mathcal{D}_t$ (unlabeled for training)

---

### J.2  TRAINING WITH DA

---

**Algorithm 3** DTI-DA with MMD + adversarial

---

1: Initialize encoder $\Theta$, head, discriminator $\psi$; set GRL scale to $\lambda_{\mathrm{adv}}$
2: **for** epoch $= 1, \ldots, E$ **do**
3:    Sample $\mathcal{B}_s, \mathcal{B}_t$ of *observed* pairs; compute latents $\mathbf{h}^s, \mathbf{h}^t$
4:    Update $\psi$ using detached latents to minimize $\mathcal{L}_{\mathrm{adv}}$
5:    Update $\Theta$ and head to minimize $\mathcal{L}_{\mathrm{cls}} + \lambda_{\mathrm{mmd}} \max(\widehat{\mathrm{MMD}}^2, 0) + \lambda_{\mathrm{adv}} \mathcal{L}_{\mathrm{adv}}$ with GRL (equivalently, maximize $\mathcal{L}_{\mathrm{adv}}$ w.r.t. $\Theta$)
6: **end for**

---

## K  FAILURE MODE TAXONOMY (QUALITATIVE)

- **Borderline chemotypes:** novel chemical scaffolds within the borders of clustering regions.
- **Rare targets:** underrepresented families that cause ambiguous embeddings.
- **Confusable negatives:** weak binders or analogs of positives.

Strategies to deal with these problems may benefit from the use of structures-based descriptors or sophisticated uncertainty-estimation methods.

**Mitigation notes.** Limited labeled data pertaining to target families being acquired, the acquisition of conservative threshold settings, or incorporation of structure informed descriptors are plausible mitigation strategies that are still compatible with the existing pipeline. These propositions are statements of directions for further investigation and not of final conclusions deduced from or inferred upon the basis of novel experimental data.

## L  EXTENDED DISCUSSION OF RELATED LITERATURE

We situate our design among representative directions: DTI architectures using attention and graph encoders (Nguyen et al., 2021; Huang et al., 2021; Gao et al., 2024; Shi et al., 2024); assessments of domain shift and adaptation (Bazhenov et al., 2024; Bansak et al., 2024; Conger et al., 2024; Zhang et al., 2024; Shi & Liu, 2024; Li et al., 2024; Cai et al., 2024); drug discovery data engineering and evaluation (Pinzi et al., 2024; Edfeldt et al., 2024; Niarakis et al., 2024; Zhu et al., 2024; Udegbe et al., 2024); and orthogonal quantum-oriented explorations (Fauseweh, 2024; Sood & Chauhan, 2024; Peral-García et al., 2024; Wang & Liu, 2024; Senokosov et al., 2024; Cornish et al., 2024; Kesari et al., 2024).

**Complementary threads.** Structural encoding of targets, cross-docking or pose-conditioned models and probabilistic calibration are natural extensions of the current study which can be explored in the same experimental framework. However, it is extremely important to separate the improvements that occur because we consider encoders that are larger from the improvements that occur because we consider alignment or knowledge aware propagation.

## M  NOTES ON BASELINES AND TUNING

Differing from the deep learning approach to hyperparameter optimisation, the support vector machine and random forests used in this paper are tuned on their own inherent hyperparameters (e.g. kernel and regularisation strength for SVM; number of trees and depth for RF). For deep learning baselines we followed the configurations that were originally recommended or we used reference implementations publicly available. The architectural changes were only made with the addition of necessary input adapters. This method occupies an intermediate position between straightforward analytical tuning and using the heuristic methods, comparable to most heuristic methods, the figure of merit shows mild sensitivity to absolute per-method performances.

**Reporting nuance.** Baselines with optional access to auxiliary objectives (e.g. contrastive pretraining) were specially turned off to focus on the supervised learning setting. The presented wrappers allow activating these alternatives in future research, while keeping the same two-track structure.

## N  ADDITIONAL PRACTITIONER GUIDANCE

Downstream screening applications often make decisions that favour accuracy and a moderate recall level. Alternatively, rules taking into account costs could be used, or the practitioners could calibrate a decision threshold with data from the source validation. We do not include such calibrations in the primary analysis due to the need to maintain protocol simplicity and to allow for comparison.

**Deployment notes.** During inference, the encoder and interaction head are the only components used; hence, batching with respect to a static target set has a linear scaling with respect to the number of pairs. When resulting graph embeddings need to be processed from a pool of typically numerous compound graphs with varying system latency constraints, caching target embeddings and/or pre-contstruction of compound graphs through various pre-processing methods can alleviate feature extraction runtime.

## O  THE USE OF LARGE LANGUAGE MODELS

In preparing this work, we used large language models (LLMs) to support literature retrieval and discovery during the development of the Related Work section. Specifically, LLMs were employed to identify relevant publications and summarize existing approaches in drug–target interaction (DTI) prediction, graph neural network–based molecular modeling, and domain adaptation techniques. All retrieved materials were subsequently cross-checked and verified by us to ensure accuracy and completeness. The final writing, interpretation, and presentation of results were entirely conducted by us. Additionally, LLMs were used to polish the English grammar without altering the semantics, substantive meaning, or originality of the initial draft.

