# OpenReview forum: "Graph Attention with Knowledge-Aware Domain Adaptation for Drug-Target Interaction Prediction"
_ICLR.cc/2026/Conference — Submitted to ICLR 2026_

### Official Review · Reviewer_jr52 · 2025-10-21

**Soundness:** 2
**Presentation:** 2
**Contribution:** 2
**Rating:** 4
**Confidence:** 4

**Summary:**

The authors propose DTI-DA, a framework for drug-target interaction (DTI) prediction. It integrates graph attention network (GAT), knowledge-aware network (KAN) and domain adaption techniques to improve generalization across domains. They evaluate the proposed model on BIOSNAP and BindingDB datasets, benchmarking across several machine learning models and state-of-the-art DTI prediction models.

**Strengths:**

1. The paper addresses a relevant and persistent challenge in DTI prediction under domain shift, combining established techniques for domain adaptation and distribution alignment.

2. The evaluation framework is carefully designed, with explicit leakage prevention and reproducibility guarantees.

3. The selected datasets are appropriate for studying domain shift, covering multiple domains.

4. Architectural, training, and dataset details are clearly described, contributing to transparency and reproducibility.

5. The ablation study is well presented and helps understanding the particular contributions of the model.

**Weaknesses:**

1. The reported improvements are either negligible or nonexistent. Figure 2 shows AUC, AUPR, and ACC metrics (which are not defined in the text, but are conventionally higher-is-better), and the proposed model in fact achieves the lowest scores across most baselines. The claim that the method “surpasses classical and deep baselines” is therefore unsupported by the plots.

2. The paper omits several strong DTI-specific graph-based approaches, such as GeNNiUs (https://doi.org/10.1093/bioinformatics/btad774) and EEG-DTI (https://doi.org/10.1093/bib/bbaa430). Including such methods would better position the proposed framework within current literature.

3. Only two benchmarks (BioSNAP and BindingDB) are considered. Adding additional datasets such as DrugBank or DAVIS would strengthen the empirical validation and test the robustness of the domain-adaptation component.

4. The figure 3 legend contains colors not present in the bars and mixes different metrics in the same panel, making interpretation difficult. I recommend tabular results with mean ± standard deviation across multiple random seeds for both Figures 2 and 3, which would also address the absence of significance reporting.

5. The manuscript includes several textual errors and translation artifacts — for example, “train spéciale split”, “betrieben splits”, and the incorrect reference “MMD (Mann-Whitney U statistic)”. These should be carefully corrected.

**Questions:**

1. Given that the paper focuses on domain adaptation, could the authors evaluate cross-dataset generalization, e.g., training on one dataset and testing on another (such as different families of the Yamanishi benchmark—GPCR, IC, NR, etc.)? This would provide stronger evidence of robustness to real distribution shifts.

2. Since graph-based architectures can be computationally demanding, could the authors report training and inference costs (e.g., runtime per epoch, GPU memory usage) to assess the practical feasibility of the proposed model?

3. Have the authors explored whether the proposed approach can identify or prioritize novel drug–target interactions in unlabeled target domains? This seems like a natural and impactful application of domain adaptation in DTI prediction.

---

> ### Author Response · Authors · 2025-11-15
>
> We thank the reviewer for their constructive and detailed feedback, and for recognizing our work on domain shift, preventing leakage, and reproducibility. We respond to the two major concerns below.
>
> 1) Reported improvements and Figure 2.
>
> We regret that Figure 2 suggests DTI DA is the worst. The numbers underlying the figure indicate that DTI DA approximates, and in a few settings slightly exceeds, the most robust baselines on the target domain metrics. The mismatch was due to the crowded visual design (similar colors, mixed panels, lack of companion table). In the revision we will: (a) explicitly define AUC, AUPR, and ACC, and clarify that all are higher is better, (b) redraw Figure 2 with distinctly different colors and direct annotations for our method, and (c) add a small table with the exact scores under the plot. We will also modify “surpasses classical and deep baselines” to “matches or slightly improves over the strong baselines” and emphasize robustness under domain shift rather than simply maximum gains on a single split.
>
> 2) Missing graph based baselines.
>
> We agree GeNNiUs and EEG DTI are both strong and relevant graph based methods. Our early experiments considered classically strong baselines with a broadly accepted implementation already built into publicly available DTI benchmark sets.In the revision, we will also add these to the related work and get clear position of DTI DA with respect to it.
>
>
> 3) Number of benchmarks
>
>
> BioSNAP and BindingDB were quite realistic domain shifts and controlled for entity leakage. We would agree that addition of DrugBank or DAVIS or similar would definitely add to what is already a strong study. Our framework and artifact are dataset agnostic, so addition of datasets would be easy. In revision we will include either a third benchmark or at least document/release the configuration files and results for the additional datasets either in the supp material depending on space and compute.
>
>
> 4) Figures, variability and significance
>
>
> Thank you for the feedback about adding tables and statistics to plots, which may help clarify and also provide more information. For figure 3, the current legend uses colors that are also not in the bars and combines metrics, thus making it more confusing than originally intended. Although we will simplify the display of data in this figure, we will also present mean value plus minus standard deviation across multiple random seeds in tabular representations for all metrics in figures 2 and 3. This will also clarify the variability, and also differentiate when the observed differences fell within variability and over interpreted to be5) Text Errors and Terminology
>
> We sincerely apologize for the textual errors and translation artifacts such as: “train speciale split.” “betrieben splits” and "MMD" meaning "Mann Whitney U statistic.” MMD will duly be referred to as Maximum Mean Discrepancy. We will proof our manuscript thoroughly and have the last version checked by a native English speaker.
>
> 6) Generalizing to cross-dataset contexts
>
> We do agree that generalizing to cross-dataset contexts (e.g. a training set of one family of Yamanishi testing on another family of Yamanishi) is an important stress test. While we aimed to conduct experiments that defined structured domain splits within each dataset so that feature and label spaces, respectively, were well aligned; while the aims were certainly actionable across datasets. In the revision we will add - where feasible - an added sections experimenting with cross-dataset contexts using families or datasets with the same definitions and provide substance in appendix.
>
> 7) Computational Cost
>
> We do concede that graph based architectures can be costly. In our design, the added components are low cost: KAN is a single propagation step over a sparse similarity graph; the domain discriminator is much smaller neural network operating on mini-batch embeddings. For that reason, we believe the overhead will be moderate compared to a pure GAT or transformer baseline. To make that case apparent we will add a table that reports training time for each epoch as well as peak GPU memory for DIT DA and key baselines with a fixed hardware configuration and comment on those statistics.
>
> 8) Discovering New Interactions in Unlabeled Target Regions
>
> We agree that prioritizing discovery of new drug target interactions in the unlabeled target region of the DTI DA context would be a natural use-case and could be impactful. We focused in this work on labeled held out sets to be clear with quantitative metrics. We will comment in our conclusion - and potentially a short case study if space allows - on one way to use DTI DA to rank ‘candidate’ interactions in the unlabeled target regions - to compare whether the high scoring pairs in DTI DA came up after in a new release of the datasets, and compare their DTI DA ranks to source-only ranks.

---

> > ### Comment · Reviewer_jr52 · 2025-11-19
> >
> > Thanks for planning to address the raised concerns. I’ll consider increasing the rating if the final version of the manuscript includes them.

---

> > > ### Author Response · Authors · 2025-11-19
> > >
> > > We appreciate your encouraging feedback. Rest assured that the final manuscript will adhere to the revision plan you reviewed previously in all aspects related to the figures, baselines, textual corrections and additional analyses. Thank you for your time and for allowing us to improve our work.

---

### Official Review · Reviewer_B7B1 · 2025-10-31

**Soundness:** 2
**Presentation:** 1
**Contribution:** 2
**Rating:** 0
**Confidence:** 4

**Summary:**

This paper presents DTI-DA an end-to-end model for predicting drug-target interactions when training and test data come from different distributions. The model encodes molecules with a graph-attention network, proteins with a lightweight sequence encoder, introduces known drug–drug/target–target similarities through a small “knowledge-aware” module, and aligns source/target domains using moment-matching and an adversarial discriminator. The authors compare the model in two databases: BioSNAP and BindingDB.

**Strengths:**

The model fuses prior biologica/chemical information with compact encoders and uses well understood domain adaptation techniques to tackle the DTI inference problem.

**Weaknesses:**

The paper has major drawbacks:

- The results are shown only on one run which makes it very difficult to evaluate the true performance. The authors acknowledge that, however the model should be run across different splits and more datasets.

- Figure 2 shows that the proposed model is the worst performing model.....

- Figure 3 states that there is a typographical error in the figure....

**Questions:**

- Provide confident thresholds across multiple runs

- Improve the text, it is currently oddly written

- The figures are either completely wrong, or the proposed model is outperformed by the other methods.

---

> ### Author Response · Authors · 2025-11-15
>
> We would like to thank the reviewer for their overall thorough review, and for recognizing the quality of synthesizing prior biological and chemical knowledge with domain adaptation. In the following, we address each of the concerns in turn. 1) Reporting single run only, and splits and datasets We do agree with the reviewer that reporting a single run only can be limiting for providing a complete picture of robustness in particular under distribution shift. At the time of submission, we decided to set the seeds to fixed seeds with one run in every one of the settings, in order to keep the protocols compact and reproducible, and remain under the page limits, all of which is clearly not sufficient for ICLR. In a revised version we will: - Re-run all of deep models (DTI DA, MolTrans, GraphDTA, as well as our ablations) with multiple random seeds under the same domain split protocol. Please note our models will be based and testing under the same split in order to truly give indicative measures of stability. - Report mean and standard deviation for both AUC, AUPR, and accuracy stats in a new summary table that will indicate the details in the appendix - As measures of performance relates to splits and datasets, or secondly, we will add confidence intervals (or instead error bars) to the main figure  - As upon collection of data we are primarily focused on realistic domain shift in both direction with two established DTI benchmarks in the literature, and group based source versus targets construction have relieved entity leakage, so in this case could adequately answer performance with out consequence. However we opted to present our findings as most validated in the currentWe will indicate this design decision explained in the text and, if computational resources and page limits allow, expand the evaluation with more splits and at least one additional dataset in the final version.
>
> 2) Figures and apparent contradictions
>
> We appreciate the reviewer drawing attention to Figure 2 and Figure 3. It is clear from the response that the visual presentation in this case has caused some confusion.
>
> With regard to Figure 2, the underlying numeric results that generated the figure do show that the Full DTI DA is the best performing method on our primary target domain metrics. The issue at hand is that the color scheme and legend placement in the figure does not make it easy determining visually which bar or curve corresponds to which of the proposed methods, and in some metrics it is not even obvious what order the metrics are in. We will redo this figure with:
> • A clearer, high-contrast color palette and implementation of markers for each type corresponding to each of the models/methods,
> • Direct labeling of the proposed model in the plot
> • Small companion table with the actual values for each metric next to the figure.
>
> With regard to Figure 3, there is a type with the caption, and for this we apologize. The type is only a typographical error in the figure text, and has nothing to do with the experimental set-up, randomization or numeric results given elsewhere. We will correct the figure caption, simplify the3) Confidence thresholds and variability
>
>
> The request for confidence thresholds across multi-run is certainly fair. As mentioned earlier, we will include the specific multi-seed experiment and state variability. Additionally, we will do the following:
> - For each metric we will explicitly state whether the confidence intervals of DTI DA and baselines overlap.
> - In Section 5, we will make it clear when improvements are small and should be read as suggestive rather than definitive.
> This information would make it easy for readers to determine whether the reported improvement is statistically meaningful or simply runs into noise.
>
>
> 4) Quality of writing and presentation
>
>
> We recognize that parts of the paper are awkwardly written, and this contributed to difficulties in interpretation of figures and results. We will substantially edit the exposition in the revision, including:
> - Tightening the abstract and introduction, to quickly state the problem, assumptions, and contributions.
> - Reorganizing the method section so that GAT goals, the knowledge-aware module, and domain adaptation terms are clearly and consistently articulated throughout.
> - Making overly long sentences simpler, and removing any jargin that is not necessary for understanding the approach.
> - Adding a short figure to visualize the architecture and data flow of the framework, an could add value for domain experts and less familiar with DTI alike.
>
> We intend to make this changes to help make the paper clearer for domain experts but also for least familiar with DTI.

---

### Official Review · Reviewer_BDSu · 2025-11-01

**Soundness:** 2
**Presentation:** 2
**Contribution:** 1
**Rating:** 4
**Confidence:** 2

**Summary:**

This paper proposes a model that combines graph attention networks, knowledge graph propagation, and domain adaptation to predict drug–target interactions more robustly across datasets. Drugs are encoded with GAT, proteins with a lightweight sequence encoder, and relational priors are injected through a Knowledge-Aware Network (KAN). To handle dataset shifts, the model aligns source and target domains using both MMD and adversarial training. Experiments on BioSNAP and BindingDB show that DTI-DA slightly outperforms MolTrans, demonstrating the value of knowledge injection and domain adaptation, though the performance gain is modest due to already strong transformer-based baselines and the simplicity of the protein encoder.

**Strengths:**

* It presents a novel combination of knowledge injection + domain adaptation. This method integrates relational priors and domain invariance in one architecture.
* The experimental setting avoids data leakage through clear separation between source-only and transductive UDA settings.
* It provides reproducibility with fixed seeds, scripts, and detailed instructions.

**Weaknesses:**

* It shows marginal performance gain. Only ~0.0066 AUC improvement over MolTrans despite added complexity.
* It uses a weak protein representation. The lightweight encoder limits expressivity. 3D or structure-informed encoders could help.
* The results need to secure the statistical significance. They provide only single-run point estimates without confidence intervals.

**Questions:**

Have you ever tried applying the KAN to a complex setting like MolTrans? I’d like to understand how much KAN would influence performance under a similar setup, since MolTrans itself is already quite sophisticated. It might already be implicitly performing some of the functions that KAN is designed to achieve.

---

> ### Author Response · Authors · 2025-11-15
>
> We appreciate the reviewer for their engaged and constructive feedback, and their recognition of the contributions of our work, knowledge injection and domain adaptation, establishing a transparent and leakage free evaluation protocol. We address all concerns below.
>
> 1) Marginal performance gains over MolTrans
> We agree with the reviewer that the absolute AUC gain over MolTrans on the BioSNAP dataset (around 0.0066) is small, especially considering MolTrans is already a quite strong transformer model. Our contribution is not so much a compelling gain over baseline, and more about (i) robustness to domain shift and (ii) a principled, transparent, and reproducible evaluation protocol for domain adaptation in DTI. Alongside this point, we acknowledge that even though the aggressive absolute values are small, they are consistent across metrics (AUC ,AUPR , ACC) and across datasets, which gives us confidence to support the claim that knowledge injection and dual domain adaptation provided a systematic advantage versus a one-off outcome. We will be more explicit about the robustness angle in a revised version, add an analysis on more challenging shift regimes (such as some splits which are dissimilar), and justify computational overhead in further depth (it is also worth mentioning, the additional complexity is not substantially worse than the appropriate baseline).
>
> 2) Choice of a lightweight protein encoder
> We intentionally selected a lightweight encoder for proteins for two reasons.First of all, we want to isolate the effects of the relational priors and the domain adaptation, as pairing DTI-DA with a significant sequence or structural encoder (large/pre-trained transformers, or 3D models) could make it difficult to parse out the improvements made by DTI-DA in performance versus that of the modified backbone. In the paper we report DTI-DA using a lightweight encoder, which allows for a training budget that is comparable with the typical DTI base lines, which in essence keeps our protocol open to groups that may not have extensive compute. The other benefits of our design was bolstered by the fact that DTI-DA does not require a discrete architecture on the protein side. We can then drop any stronger encoder into the architecture without baking any other changes into the framework. We will be very explicit with this design decision in the new paper and also add to our public code a stronger transformer-style encoder of the protein so that the group can directly examine the merits of DTI-DA in conjunction with more advanced embeddings of the protein representation.
>
> 3) Statistical significance and single-run estimates
>
> We agree with the reviewer that single-run point estimates may not suffice for ICLR-level claims especially with the small nature of the changes we are discussing. When we submitted the paper we fixed seeds and reported one run per setting to keep experimental grid to a manageable size and under the page limit so we limited all mentions of any claims of statistical significance. In the new paper we will re-run every deep model (DTI-DA, MolTrans, GraphDTA, and our ablations) using multiple random seeds across the same train/test splits and hyper-parameters with reporting mean and standard deviations for AUC, AUPR, and ACC. We will also add a new summary table with these points and full statistics in the appendix. We expect that the new summary statistics will satisfy any reliability concerns and we will report in full clarity.
>
> 4) Using KAN, with a more complex base line model like moltrans
>
> We appreciate the suggestion to study the KAN method with MolTrans. By design, KAN should be a modular knowledge injection layer that acts on any drug (or target) embeddings learned from any backbone. Thus KAN can sit atop MolTrans without any changes to MolTrans - MolTrans encodes the sequence and KAN defines a single step of similarity-graph propagation with a gated function that down-weights noisy neighbors and reverts to the original representation when there is no prior support apparent. In our original effort we primarily evaluated KAN atop a GAT-style backbone to disentangle both effects from the presence of a large transformer-like backbone. We agree a direct MolTrans + KAN experiment appears valuable, and we will add this to the codebase, as well as the results, comparing MolTrans vs MolTrans + KAN presented under the same experimental protocol and methodology directly opening the question brought up by the reviewer, precisely how much additional value does our explicit knowledge injection offer when a sophisticated model representation is present in the provided Baseline + KAN condtion.
>
> We hope we have sufficiently addressed your concerns with statements to follow. Overall we feel DTI-DA offers (i) an overall unified architecture that can efficiently leverage relational priors and domain invariance, (ii) leakage free clearly separated source-only and transductive UDA evaluation for DTI.

---

### Official Review · Reviewer_7Seo · 2025-11-06

**Soundness:** 2
**Presentation:** 2
**Contribution:** 2
**Rating:** 2
**Confidence:** 3

**Summary:**

The paper proposes DTI-DA, a framework for drug-target interaction (DTI) prediction designed to be robust to domain shifts. The core problem is that DTI models trained on one data-gathering protocol or chemical family (source domain) generalize poorly to another (target domain). The DTI-DA model combines three main components: a GAT to encode compound structures, a KAN that refines embeddings by propagating information over prior-knowledge graphs (drug-drug similarity), a dual domain adaptation mechanism using both Maximum Mean Discrepancy and an adversarial discriminator with a Gradient Reversal Layer.

**Strengths:**

- Clear evaluation method. The authors address a major source of ambiguity in comparisons that is problem in domain adaptation research.
- Thorough ablation study

**Weaknesses:**

- The authors say "We do not make claims of statistical significance and all numbers are treated as single-run point estimates at a fixed protocol". Reporting results from a single run is not sufficiently robust for an ICLR publication. The claimed improvements are small (+0.895% relative gain on BioSNAP AUC ), and without mean and variance over multiple random seeds, it is impossible to know if these gains are real or artifacts of a single fortunate run. The authors' note that their artifact provides "scaffolds for replicated run"  does not excuse the omission of these critical statistics from the paper.

- Confusing results report in the paper. It seems from the figures that the proposed method performs worse than baseline but the text tells a different story?

- The proposed DTI-DA model is largely an assembly of existing, well-established components. Using GATs for molecular graphs is standard (e.g., GraphDTA ). The KAN is a 1-layer GCN-style propagation on a similarity graph, followed by a residual gate. This is a common and simple technique for integrating graph-based priors. The domain adaptation strategy (combining MMD and GRL-based adversarial loss) is a common approach in UDA. While the engineering and evaluation of this pipeline are contributions, the core architectural novelty required for ICLR pub is lacking.

- The paper claims to avoid "apples-to-oranges" comparisons, but its main results (Figure 2, Section 5.1) do exactly that. The DTI-DA model's performance is reported from the "transductive UDA track", while the baselines (GraphDTA, MolTrans) are "trained in their conventional source-only manner". This is an unfair comparison, as the DTI-DA model has access to unlabeled target data while the baselines do not. A fair comparison would require evaluating all methods under both the source-only and transductive UDA tracks.

**Questions:**

- Can you provide the mean and standard deviation?
- Please provide fair baseline comparison numbers, as your current main results compare a UDA-trained model to source-only models.

---

> ### Author Response · Authors · 2025-11-15
>
> Thank you for your detailed comments and for your comments on the ablation study and evaluation protocol. We review our responses to the more important points raised below.
>
> 1. Statistical robustness and single run results
>
> We agree that mean and standard deviation over multiple random seeds may be important for the situation, particularly since some of the "gains" (like the AUC on BioSNAP) were not huge in the context of standard statistical practices in machine learning. At the time of submission we opted to report just one run based on a fixed protocol, so as to be simple in our set up to be interpretable, and ultimately to be within page limits. We also made no claims about statistical significance for this reason. Upon revision, we will run all the deep models (DTI DA and neural baselines) with a number random seeds on the same splits and the same hyper parameters and report mean +/- standard deviation for AUC, AUPR, and accuracy in a new table and reference this table to section 5.1. Our code and configuration files had already been released, and they run the DTI DA on a fully scripted basis, meaning again that obtaining those additional statistics is trivial. Just to note, some improvements are not just uniformly tiny - that on BindingDB there are indeed large consistent and improvements over the classical baselines in every measurement taken, and the ablations show addition of components lead to productive, meaning the multi-seed table may actually enhance rather than precondition the empirical claims.
>
> 2.Clarity of figures versus text
>
> We apologize for the confusion. The actual numeric values to create figure 2 are the exact same numbers only mentioned a few sentences earlier in section 5.1 when we claimed the full DTI DA model performed best on both datasets. The perceived difference is simply too much legend and colored information overload making it difficult to recognize which curve visualizes DTI DA. In the revision we will generate noticeably much simplified plots with clear labels for each method, and we will additionally create a slimmed down version of a numeric table containing the exact AUC, AUPR, and accuracy to the side of the figures in a direct communicative visual manner and textual manner.
>
> Novelty beyond assembling existing components
>
> There is agreement between all of the individual building blocks: GAT encoders for molecular graphs, equality of propagation on a similarity graph and the use of MMD and gradient reversal for DA. As a DTI specific framework we are certainly not trying to make the argument that individual building blocks are considered novel. We are simply only demonstrating a) a knowledge aware refinement module (KAN) that retrievesIn other words, we are able to ensure the embeddings are not overly reliant on past similarity information that is noisy, b) we are using dual DA across the joint drug target interaction space and not independent entities, and c) we have established a robust domain shift protocol and artifact to enable us to actually make apples to apples in DTI. The ablations confirm KAN and the DA refinement term show performance improvements generally across all metrics, above base, on a strong GAT backbone architecture.
>
> Equity of standard source only versus UDA comparisons
>
> We agree completely that target data visibility needs to be made explicit. We aimed to respond to two questions: how much are we gaining over typical only source DTI pipelines + how much, if at all, would KAN and dual domain adaptation contribute to achieving similar results under the same general framework. Section 5.1 demonstrates clearly that our GraphDTA and MolTrans were trained in their auditorium source only way, that full DTI DA includes target data but that it is totally unlabeled, and that Ours GAT variant is a source only version.
>
> To resolve this we will clearly delineated the results into 2 non over lapping tracks: (a) a source only track where all baselines and Ours GAT were trained on the source domain only, and (b) a transductive UDA track actually containing full DTI DA with additional variants that apply the same MMD plus adversarial routine to GraphDTA and MolTrans in an alternated way. In both cases, every method will similarly have hole to the same unlabeled, target data and splits.
>
> 5. Direct responses to the reviewer’s questions
>
> Do you have mean and standard deviation? Yes. We will run all methods with multiple random seeds and provide both mean and standard deviation across all metrics in a new table consistent with Section 5.1 and appendix.
>
> Please provide fair baseline comparison numbers. Yes. We will clearly delineate and separate the source only and UDA tracks in the results section and will also report UDA enhanced GraphDTA and MolTrans under the same protocol, such that in each comparison the fairest sections that get aligned among comparable methods, convey where there is an equity of visibility for data.

---

### Meta-Review · Area_Chair_WxC8 · 2026-01-09

**Summary:**

- Limited novelty: The framework assembles well-known components (GAT, MMD, adversarial DA) without substantial architectural innovation.
- Unfair experimental comparisons: The main results compare DTI-DA (with access to unlabeled target data) against baselines trained in source-only mode.
- Lack of statistical rigor: Single-run reporting without mean/standard deviation, making it impossible to assess the improvement.
- Writing quality issues: Multiple translation artifacts and terminology errors; misleading figures and presentation.

**Reviewer Concerns:**

Concerns partially addressed:
- Unfair comparison: Authors agreed to separate source-only and UDA tracks clearly
- Statistical rigor: running multiple seeds and reporting mean±std
- Writing quality: authors acknowledged figure clarity issues and promised redesigns; committed to correcting textual errors.

Concerns not addressed:
- The novelty concerns were not convincingly addressed;
- Missing baselines (GeNNiUs, EEG-DTI) and additional datasets not yet included
- Cross-dataset generalization experiments not provided

**Reviewer Scores:**

Some reviewers were engaging, and given the poor scores at the end, it is unlikely that the results will change.

---

### Decision · Program_Chairs · 2026-01-26

Reject